# Genome-Wide Identification of the Brassinosteroid Signal Kinase Gene Family and Its Profiling under Salinity Stress

**DOI:** 10.3390/ijms25158499

**Published:** 2024-08-04

**Authors:** Biao Shi, Youwu Wang, Liang Wang, Shengwei Zhu

**Affiliations:** 1College of Agriculture, Tarim University, Alar 843300, China; suifengwuyu@outlook.com (B.S.); wangyw1975@126.com (Y.W.); 2CAS Engineering Laboratory for Grass-Based Livestock Husbandry, Institute of Botany, Chinese Academy of Sciences, Beijing 100093, China; 3Biotechnology Research Institute, Xinjiang Academy of Agricultural and Reclamation Science, Shihezi 832000, China

**Keywords:** alfalfa (*Medicago* L.), BSK gene, salinity stress, expression pattern

## Abstract

Alfalfa (*Medicago* L.) is a high-quality perennial leguminous forage with the advantages of salt tolerance, mowing tolerance, high protein content, and other economically valuable characteristics. As the sixth class of plant hormones, brassinosteroids (BRs) play indispensable roles in modulating a variety of plant growth, maturation, and environmental adaptation processes, thereby influencing vegetal expansion and development. Brassinosteroid signal kinases (BSKs) are key cytoplasmic receptor kinases downstream of the BR signaling transduction pathway, participating in plant growth, development, and stress regulation. However, the phylogenetic and expression pattern analyses of the BSK gene family among the five alfalfa species have rarely been reported; in this study, 52 BSK family members were found in the genomes of the five subspecies, and phylogenetic trees were constructed according to protein sequences, allowing us to categorize all BSKs into seven distinct groups. Domain, conserved motif, and exon–intron structural analyses showed that most BSK members were relatively conserved, except for MtBSK3-2, MtBSK7-1, and MtBSK7-2, which may be truncated members. Intra-species collinearity and Ka/Ks analyses showed that purifying selection influenced BSK genes during evolution; most of the cis-acting elements in the promoter region were associated with responses, such as light, defense, and stress, anaerobic induction, MeJA, and abscisic acid. Expression pattern analysis indicated that the majority of alfalfa genes exhibited downregulation after reaching a peak at 0.5 h after treatment with 250 mM NaCl, especially for MsBSK14, MsBSK15, MsBSK17, MsBSK19, and MsBSK21; meanwhile, MsBSK4, MsBSK7, and MsBSK9 increased and were highly expressed at 12 h, demonstrating significantly altered expression patterns under salt stress; furthermore, MsBSK4, MsBSK7, and MsBSK9 exhibited expression specifically in the leaves. qRT-PCR analysis confirmed the expression trends for MsBSK4, MsBSK7, MsBSK9, MsBSK14, MsBSK15, and MsBSK16 matched the transcriptome data. However, the trends for MsBSK17, MsBSK19, and MsBSK21 diverged from the transcriptome data. Our study may provide a foundation for further functional analyses of BSK genes in growth, development, and salt stress tolerance in alfalfa.

## 1. Introduction

Brassinosteroids (BRs), recognized as the sixth major class of plant hormones, play crucial roles in various aspects of plant growth and development, including seed germination, cell elongation, photomorphogenesis, xylem differentiation, and reproduction [1]. Not only do they contribute to normal plant development, but they also assist in resisting high-salinity and alkaline conditions. The exogenous application of BRs has been shown to bolster seed resistance to salinity stress [2,3,4,5,6], while also alleviating the harmful effects of saline conditions and alkali on plant health, according to studies [7,8]. BR signal kinase (BSK) is an important cytoplasmic receptor kinase downstream of the BR signal transduction pathway that serves as a substrate of the brassinosteroid receptor BRI [9] and plays a crucial role in regulating plant growth, development, and abiotic stress responses. Originally, members identified within this gene family, *AtBSK1* and *AtBSK2*, were established as BR response proteins [10], as revealed by binary differential gel electrophoresis and mass spectrometry. The typical structures of BSK gene family members are N-terminal kinase domains and C-terminal tetrapeptide repeats. Of the twelve BSK members in *Arabidopsis thaliana*, the second serine homologous checkpoint (S^230^ in *AtBSK1* and S^210^ in *AtBSK3*) has been reported via AtBRI1 as a phosphorylation site in the BR signaling pathway [9,11]; the second glycine is a highly conserved N-myristoylation site in BSKs [12]; and alfalfa BSK family proteins all have at least one S-palmitoylation sites and multiple ubiquitination sites [13,14].

Alfalfa (*Medicago* L.) is one of the most widely distributed forages, holding great significance for the development of animal husbandry. Alfalfa has high feed value, great production potential, a developed root system, and rhizome buds with super tillering capacity; it is the preferred forage for restoring grasslands for ecological protection and supplying higher livestock biomass, and it plays a crucial role in agricultural production as well as ecological protection. High-salt environments induce osmotic stress and ion toxicity in alfalfa, leading to stunted growth and reduced biomass. Consequently, understanding the mechanisms of salt tolerance in alfalfa and developing effective salt tolerance strategies are essential to mitigate the adverse effects of salinity stress on this crop. With the release of the genome of alfalfa plants [15,16,17,18,19], there have been more and more reports on gene families, such as the *MYB* [20], *WRKY* [21,22], *SAUR* [23], *WOX* [24,25], *CAMTA* [26], *BGLU* [27], *MADS*-box [28,29], *BBX* [30], *B3* [31], *CCoAOMT* [32], *TCP* [33], and *ERF* families [34,35]. Current studies on the BSK gene family are mainly conducted on Arabidopsis [13], cotton [36], rice [37], spinach [38], maize [39], potato [40], etc.; meanwhile, in alfalfa, the BSK gene family has never been identified and analyzed. In this study, based on bioinformatic methods conducted on five sequenced alfalfa genera (*Medicago* L.), 52 BSK members were identified among these subspecies, and their physicochemical properties, subcellular localization prediction, chromosome localization, phylogenetic trees, gene structures, and promoter-localized cis-acting regulatory elements were predicted and analyzed for intra-species collinearity. Building on previous findings, the post-treatment expression and qRT-PCR data of alfalfa were further explored, providing theoretical insights into the function of BSK genes in response to salinity stress.

## 2. Results

### 2.1. Identification and Physicochemical Characteristic Analysis of the BSK Family

In total, 52 BSK genes were discovered in alfalfa (Table 1), including 24 MsBSK, 9 MtBSK, 6 MpBSK, 7 MrBSK, and 6 MsaBSK genes, which were named according to their distributions on their respective genomic chromosomes. There were 353–526 amino acids encoded by the BSK family in five alfalfa species, among which the MtBSK7-1 gene encoded the fewest amino acids (353 aa) and MsBSK12 encoded the most amino acids (526 aa); their molecular weights (MWs) ranged from 39,466.97 kDa to 59,700.39 kDa; and their theoretical isoelectric point (pI) difference is small, ranging from 5.3 to 6.69. The hydrophobicity index (GRAVY) of every alfalfa BSK family protein was less than 0, while the aliphatic index (AI) was less than 100, and the instability coefficient (Ins) was less than 40 for unstable proteins. Among the five alfalfa subspecies, the number of unstable proteins varied, including *Medicago sativa* (ten: MsBSK4, MsBSK7, MsBSK8, MsBSK9, MsBSK12, MsBSK13, MsBSK15, and MsBSK16), *Medicago truncatula* (four: MtBSK4, tBSK5, MtBSK7-1, and MtBSK7-2), *Medicago polymorpha* (three: MpBSK4, MpBSK5, and MpBSK6), *Medicago ruthenica* (one: MrBSK2), and *Medicago sativa* spp. *caerulea* (three: MsaBSK2, MsaBSK3, and MsaBSK4). Subcellular localization prediction showed that the probability of the localization of BSKs in the nucleus was the highest, followed by the cytoplasm, chloroplast, and Golgi apparatus.

### 2.2. Unraveling the Phylogenetic Relationships among BSK Genes

In order to compare the evolutionary relationships between alfalfa BSKs and homologous genes from other species, a phylogenetic tree (Figure 1) was constructed using proteins from the alfalfa BSK family and other BSK families. In addition to the 52 BSK gene family proteins in alfalfa, there are 26 other species with a total of 264 BSK family members. The BSK phylogenetic tree shows seven different evolutionary tree branches, of which Group 1 has the most members (70) and Group 5 has the fewest (5), while Groups 2, 3, 4, 6, and 7 have 46, 50, 45, 45, and 55 members, respectively. The distribution of alfalfa BSK members is relatively concentrated on the evolutionary tree. *Medicago sativa* has the largest distribution in Group 6 (seven MsBSK genes), and all other branches, except Group 5, have a wide distribution. In the phylogenetic tree, there are several pairs of genes with high sequence similarity in the Arabidopsis BSK family [13], such as AtBSK3 and AtBSK4, AtBSK7 and AtBSK8, and AtBSK9 and AtBSK10. In *Medicago sativa*, MsBSK14, MsBSK15, and MsBSK16 are closely related to AtBSK3 and AtBSK4; MsBSK4, MsBSK7, and MsBSK9 are closely related to AtBSK7 and AtBSK8; and MsBSK3, MsBSK6, MsBSK8, and MsBSK11 are closely related to AtBSK9 and AtBSK10. These three categories show different functions due to differences in protein sequences.

### 2.3. Gene Structure and Conserved Motif Analysis of the BSK Gene Family

Five BSK protein sequences from five subspecies were used to construct an evolutionary tree (Figure 2A), and the five alfalfa BSK members were divided into six groups. Conservation motif analysis showed (Figure 2B) that 15 motifs were found. Among them, motifs 1, 2, 4, 6, 7, 9, 11, and 12 are conserved motifs specific to kinase domains, and motifs 3, 5, 8, and 10 are conserved motifs from TPR domains. All family members, except for MtBSK3-2, MtBSK7-1, MtBSK7-2, and MsaBSK1, have these motifs. The most significant domain features of the BSK family are N-terminal kinase domains and C-terminal TPR domains. Among them, the PLN03088 superfamily has two TPR domains, and the 3a0801s09 superfamily has three TPR domains. Domain analysis showed (Figure 2C) that 11 proteins had no TPR domains, 28 proteins had PLN03088 superfamily domains, 9 proteins had 3a0801s09 superfamily domains, and only the MtBSK2 protein contained a TPR domain. Intron–exon structural analysis showed (Figure 2D) that alfalfa BSK family members maintained a certain diversity in their numbers and lengths of introns. The proteins MtBSK4, MpBSK6, MsBSK2, MsBSK13, MsBSKI2, MsaBSK3, MtBSK3-2, and MtBSK7-1 all have eight introns; the protein MtBSK7-2 has nine introns; and the remaining BSK family proteins each have ten introns. In addition, only 12 members have 5′ UTR and 3′ UTR.

### 2.4. Chromosome Mapping of Alfalfa BSK Gene Family Members

The members of the BSK gene family are unevenly distributed on the chromosomes; Figure 3A–E show their chromosome localization. MsBSK family members are mainly distributed on 13 chromosomes, including chr4.1-chr8.4, chr 4.1, chr4.2, and chr4.4, on which three MsBSK genes are distributed, while chr4.3 and chr8.1–8.4 have two BSK genes, and chr5.1, chr5.2, chr7.1, chr7.2, and chr7.4 have one BSK gene. MtBSK family members are mainly distributed on four chromosomes, including CM001220.2–CM001224.2, where chromosomes CM001220.2 and CM001224.2 each distribute four MtBSK genes. MpBSK family members are mainly distributed on four chromosomes, including GWHAWN00000001.1–GWHAWN00000003.1 and GWHAWN00000007.1, among which two MpBSK genes are distributed on chromosomes GWHAWN00000001.1 and GWHAWN00000002.1, and one MpBSK gene is distributed on GWHAWN00000001.1 and GWHAWN00000007.1. Members of the MrBSK family are primarily located on chromosomes Mru4, Mru5, and Mru8, as well as on unassembled contigs 1980, 2048, and 2135. Specifically, two MrBSK genes are mapped to the Mru8 chromosome, while a single MrBSK gene is found on Mru4, Mru5, and each of the unassembled contigs: 1980, 2048, and 2135. Members of the MsaBSK family are predominantly found on Chr4, Chr5, Chr7, and Chr8. Specifically, two MsaBSK genes are located on both the Chr4 and Chr8 chromosomes. Moreover, the alfalfa BSK gene exhibits tandem repeat sequences. Additionally, the gene pairs MsBSK1 and MsBSK2, MtBSK3-1 and MtBSK3-2, and MtBSK7-1 and MtBSK7-2 form tandem repeat gene clusters on various chromosomes.

### 2.5. Gene Duplication and Interspecies Collinearity of BSK in Alfalfa

In order to explore the evolutionary process of BSK genes, the collinearity of BSK gene families was analyzed in each of the five alfalfa species (as shown in Figure 4), and 49 gene pairs were found. Among them, most of the replicated gene pairs belonged to the MsBSK family (42 pairs), and a total of 23 genes were involved in replication. MsBSK14–MsBSK24 participated in the replication of four chromosomal homologous genes. The fewest replicated gene pairs were from the MtBSK family (one pair), and the MpBSK, MrBSK, and MsaBSK families had two pairs each. No tandem duplicates were found among these gene pairs, all of which were WGD or segmental, indicating that whole-gene replication was the main driving force behind their evolution.

The Ka/Ks ratio is an indicator used to evaluate the pressure of the natural selection of genes in the evolutionary process, representing the evolutionary direction of genes during a species. The Ka/Ks ratios for 49 gene pairs were computed across five subspecies of alfalfa (Figure 5). The Ka/Ks ratios of 10 gene pairs in *Medicago sativa* and 1 gene pair in *Medicago ruthenica* were all 0, a result which was caused by the absence of non-synonymous substitutions between sequences. NaN appears when calculating Ka/Ks in *Medicago truncatula*, caused by the large gap between the two sequences. The majority of gene pairs exhibit Ka/Ks ratios below 0.5, surpassing this threshold, and only two gene pairs are greater than 0.5. This observation suggests that the genes in the alfalfa genome are subject to purifying selection pressure in order to preserve their functionality.

In order to explore the evolutionary relationship between alfalfa BSK genes and BSK genes in other species, we selected *A. thaliana*, the earliest discovered species of BSK genes, and the salinity-tolerant crop *Gossypium hirsutum* for collinearity analysis among species. The interspecies collinearity results indicated that the MsBSK gene family demonstrated 49 linear pairs with the GhBSK gene family and 25 linear pairs with the *A. thaliana* gene family (Figure 6); thus, the covariance of the alfalfa BSK gene with *G. hirsutum* was significantly higher than that with *A. thaliana*. Further comparison of the homologous genes among the three species disclosed that most of them had direct homologs in *A. thaliana* and *G. hirsutum*, except for MsBSK6, MsBSK8, MsBSK12, and MsBSK15, indicating that the main cause for the amplification of BSK genes might be the duplication of the genes during the species’ evolution.

### 2.6. Analysis of BSK Gene Promoter Acting Elements

Within the BSK genes, 35 types of cis-acting elements have been identified, which are involved in different abiotic stresses, including development (nine types), environmental (six types), hormonal (eight types), light (seven types), and other factors (five types) (Figure 7); among these are elements responsive to the environment (ARE), hormones (ABRE, CGTCA-motif), and light (G-box, 3-AF1 binding site, Box 4, and TCT-motif), which constitute the major part of each BSK gene. Some elements are conserved within the BSK genes, indicating that BSK genes play a role in responding to environmental stresses, light responses, and hormonal signals. In the BSK genes, a variety of plant-hormone-related elements have been identified that are involved in defense and stress responses (36 TC-rich repeat) and anaerobiosis (127 ARE), as well as low-temperature (19 LTR), drought (30 MBS), wound (5 WUN-motif), MeJA (75 TGACG-motif), abscisic acid (90 ABRE), auxin (15 TGA-element, 4 AuxRR-core), gibberellin (26 P-box), salicylic acid (32 TCA-element), and light (14 ACE, 107 G-box, 104 GT1-motif, 42 MRE, 136 Box, 136 TCT-motif, and 40 AE-box) responses. The presence of these highly conserved elements within the promoter region underscores the significance of BSK genes in various biological processes, including abiotic stress responses, light-induced reactions, and signaling pathways mediated by plant hormones.

### 2.7. Analysis of Tissue Expression Patterns and Salt Stress Responses in M. sativa

In order to investigate the connection between alfalfa BSK genes and their response to salt stress, we leveraged existing transcriptome sequencing data to examine gene expression patterns in alfalfa when subjected to a 250 mmol/L NaCl challenge at intervals of 0, 0.5, 1, 3, 5, 12, and 24 h. As shown in Figure 8B, according to the expression level, the 24 genes were divided into the three following clusters: the first cluster had higher expression, the third cluster followed, and the second cluster had the lowest expression. The majority of genes displayed a downregulation trend in expression levels at 0.5 h post-treatment. In the first cluster, the expression levels of MsBSK17, MsBSK19, and MsBSK21 were the highest at 0.5 h, and only the genes MsBSK4, MsBSK7, and MsBSK9 reached their expression peaks at 12 h; in the second cluster, only MsBSK3 and MsBSK8 showed higher expression levels under control than under salt treatment conditions; and in the third cluster, only MsBSK13 reached its peak at 1 h.

BSK genes exhibit distinct expression patterns in different tissues of alfalfa, which are detailed below (Figure 8B). The expression levels of MsBSK17, MsBSK19, MsBSK21, and MsBSK23 in roots are slightly higher than those in other tissues. Combined with phylogenetic analysis, the Group 6 subfamily is likely predominantly expressed in roots. The expression abundance levels of the MsBSK3, MsBSK6, MsBSK8, MsBSK11, MsBSK18, MsBSK20, MsBSK22 and MsBSK24 genes in elongated stems were higher than those in other tissues. Compared with other organ tissues, the transcript levels of the MsBSK4, MsBSK7, and MsBSK9 genes were the highest in leaves, while the MsBSK14, MsBSK15, and MsBSK16 genes were the highest in roots, and MsBSK12 and MsBSK13 were the highest in flowers. Consistent with their evolutionary kinship, these genes’ responses to salt stress show a similar trend in different tissue types.

### 2.8. qRT-PCR under Salt Stress in M. sativa

Nine genes, including MsBSK4, MsBSK7, MsBSK9, MsBSK14, MsBSK15, MsBSK16, MsBSK17, MsBSK19, and MsBSK21, were selected for qRT-PCR (Figure 9) based on the clustering results in the phylogenic tree and the high as well as low gene expression in the transcriptome data, in which the expression levels of most of the genes were highest at 0.5 h and began to be downregulated over time, while only MsBSK4, MsBSK7, and MsBSK9 were upregulated with a longer treatment duration. In this experiment, only the expression trends in the MsBSK4, MsBSK14, MsBSK15, and MsBSK16 genes were consistent with the transcriptome data. Although the expression trends in the MsBSK7 and MsBSK9 genes were generally consistent with the transcriptome data, their expression levels at 0.5 h–12 h were lower than those of the CK group. The transcript levels of the MsBSK17, MsBSK19, and MsBSK21 genes showed downregulation followed by upregulation starting at 3 h.

## 3. Discussion

BSKs belong to RLCK-XII, they have a kinase domain at the N-terminal and two–three TPRs at the C-terminal, and are anchored to the cytoplasmic membrane at the N-terminal myristeylation site [9]. As a key protein in brassinolide signal transduction, BRI1 is activated after forming a complex with BAK1 after BR signaling, and BSKs are phosphorylated by activated BRI1, which then transmits the signal to the downstream component BUS1 [41]. BSKs are crucial for various aspects of plant physiology, including immune responses to biotic stress, tolerance to abiotic stress, and the regulation of hormonal responses. In *A. thaliana*, BSK1 participates in the EDR2 signaling pathway and also binds to FLS2 to initiate an immune response [42]. The dynamic spatial reshuffling of BSK1 in the plasma membrane forms the basis for specific activation pathways in growth and immune signaling [1]. Three Arabidopsis proteins, PAT19, PAT20, and PAT22, positively regulate BR signaling through the S-palmitoylation of BSK1 [14]. The ZmBSK1-ZmCCaMK module also plays an important role in improving maize tolerance to oxidative stress [43]. BSK3 may function as a scaffold protein that regulates BR signaling [10] and is primarily regulated by BSK3 in response to mild nitrogen deficiency during root formation [44]. The overexpression of BSK5 enhances disease resistance to *Pseudomonas syringae*; furthermore, it provides protection against *Botrytis cinerea* [45]. BR signaling is enhanced through the action of BSK5 and leads to the activation of BES1 [46]. BSK5 plays an essential role in conferring tolerance to abiotic stresses, including salt and drought conditions [47]. The protein AT5G58150 may participate in the plant’s response to salinity and osmotic stress following interactions with BSK and other related proteins [48]. ZmBSK7 is induced by salt stress, localizing to the plasma membrane, and its overexpression increases salt tolerance in maize [49]; some transcriptomic analyses suggest that salt also induces BSK gene expression [50,51,52,53].

Studies in the literature [13] indicate that the BSK family comprises 12 distinct members in *Arabidopsis thaliana*. The BSK gene family is composed of various numbers of members across different plant species [36,37,38,39,40], as follows: 7 in *Beta vulgaris*, 13 in *Gossypium arboretum*, 27 in *Gossypium barbadense*, 28 in *Gossypium hirsutum*, 15 in *Gossypium raimondii*, 13 in *Glycine max*, 18 in *Nicotiana tabacum*, 12 in *Nicotiana sylvestris*, 5 in *Oryza sativa*, 14 in *Populus trichocarpa*, 7 in *Spinacia oleracea*, 8 in *Theobroma cacao*, 7 in *Vitis vinifer*, 11 in *Zea mays*, and 7 in *Solanum tuberosum*. In addition, some BSK family members have been identified, as follows: 5 in *Sorghum bicolor*, 7 in *Melilotus albus*, 16 in *Cicer arietinum*, 13 in *Trifolium pratense*, and 14 in *Eucalyptus grandis*. In the *Arabidopsis* BSK phylogenetic tree, AtBSK3, AtBSK4, AtBSK7, and AtBSK8 belong to the same branch, which seems to play an important role in regulating BR signaling [54]. In this study, the phylogenetic tree was divided into seven groups, and no *Arabidopsis thaliana* or alfalfa BSK family members were distributed in Group 5, probably because this branch is monocotyledonous. Compared to cotton [36] and rice [37], this outcome is similar to the BSK classification used for spinach [38]. In the analysis of cis-acting elements, a substantial number of elements related to light reaction (G-box, GT1-motif, MRE, Box 4, TCT-motif, and AE-box) were identified, which is consistent with the findings of Lei [36] in cotton. In addition, there are a large number of components related to gibberellin (P-box), anaerobic induction (ARE), salicylic acid (TCA-element), zein metabolism (O2-site), MeJA (TGACG-motif), endosperm expression (GCN4_motif), abscisic acid (ABRE), defense and stress (TC-motif), and drought inducibility (MBS) reactions, which contribute significantly to growth, development, and abiotic stress responses. In 250 mM salt stress expression profiles, as shown in Figure 6A, MsBSK1, MsBSK2, MsBSK5, MsBSK10, and MsBSK17-MsBSK24 all had higher expression levels at 0.5 h, which then began to decrease. MsBSK17, MsBSK19, and MsBSK21 displayed the highest expression levels; on the other hand, those of MsBSK1, MsBSK2, MsBSK5, and MsBSK10, along with MsBSK18, MsBSK20, MsBSK22, MsBSK23, and MsBSK24, were lower under salt stress. Among all samples, MsBSK3 and MsBSK8 exhibited the highest levels of expression in the CK group, prompting our speculation that these genes potentially exhibit minimal sensitivity to saline stimuli. MsBSK6 and MsBSK11 showed upregulation again at 12 h, after treatment for 0.5 h. Although MsBSK12, MsBSK14, MsBSK15, and MsBSK16 also started to decrease after 0.5 h, MsBSK4, MsBSK7, and MsBSK9 became upregulated after salt treatment and reached their maximum values at 12 h. Combined with the previous evolutionary tree analysis, MsBSK4, MsBSK7, and MsBSK9 were closely related to AtBSK7 and AtBSK8; MsBSK14, MsBSK15, and MsBSK16 were closely related to AtBSK3 and AtBSK4; and MsBSK1, MsBSK2, MsBSK5, and MsBSK10 were closely related to AtBSK5. A similar expression pattern was also demonstrated in tissues and organs. These closely related MsBSK genes may play the same roles in salt stress, but further experiments are required to verify which genes contribute to salt tolerance. Using transcriptome data, 24 MsBSK gene expression patterns were determined, and it was observed that most of the genes were downregulated after 0.5 h of treatment when subjected to 250 mM salt stress, except for MsBSK4, MsBSK7, and MsBSK9, which were upregulated with treatment duration. In the qRT-PCR test, MsBSK7, MsBSK9, MsBSK17, MsBSK19, and MsBSK21 still differed from the transcriptome data, despite the use of the same samples and treatments, a situation that was also found in another study that used the same transcriptome data [28]. In addition to the differences between individual samples, we also believe that the transcriptome data comprise a wide dynamic range, which leads to differences in genes with extreme changes in expression, whereas qRT-PCR is more accurate in detecting low-expression genes due to its high sensitivity. Nevertheless, we still believe that both the transcriptomic data and the qRT-PCR results provide valuable insights that contribute to realizing the complexity of the regulatory mechanisms of gene expression. Research on the BSK gene family in alfalfa, particularly regarding its role in growth, development, and response to abiotic stress, has not received much attention; further efforts should be made to explore the functions of this gene family and its process of adapting to the environment, areas of great significance for improving the production performance and stress resistance of alfalfa.

## 4. Materials and Methods

### 4.1. Identification and Physicochemical Characteristic Analysis of the BSK Family in Alfalfa

In order to determine the members of the alfalfa BSK gene family, the protein sequences containing the PKc (PF07719) and TPR (PF07714) domains were retrieved via HMMER3.0, and the E-value was set to 0.001. Using TBtools-II software [55], the amino acid sequences of AtBSKs were used as seed sequences against which to compare the protein sequences of 5 alfalfa genomes (*M.sativa*, ‘XinJiangDaYe’, 2n = 4× = 32; *M. truncatula*, MedtrA17_4.0, 2n = 2x = 16; *M. polymorpha*, GWHANWO00000000.1, 2n = 2x = 14; *M. ruthenica*, ASM1820801v1, 2n = 2x = 16; *M. sativa* spp. *caerulea*, PI464715; 2n = 2x = 16), and the following screening conditions were set: identity > 35%, E-value < 1 × 10^−5^. After initial screening, the protein sequences of the candidate members were submitted to databases such as Batch CD-search (https://www.ncbi.nlm.nih.gov/Structure/bwrpsb/bwrpsb.cgi, accessed on 8 March 2024) and SMART (https://smart.embl.de/, accessed on 11 March 2024) for domain prediction. Subsequently, the genes were numbered according to their positions on the chromosome. Molecular weight, theoretical pI, the instability index, the aliphatic index, the grand average of hydropathicity, and other indicators were predicted using TBtools-II software, and their subcellular localization was predicted using WoLF PSORT (https://wolfpsort.hgc.jp/, accessed on 26 February 2024).

### 4.2. Structural Characteristics of the BSK Gene Family

Family member protein sequences were analyzed to determine domain structures using the MEME Suite (https://meme-suite.org/meme/, accessed on 7 March 2024), with the motif number set to 15. Subsequently, the gene structure was visualized with TBtools-II and refined for presentation using Adobe Illustrator 2021.

### 4.3. Construction of Evolutionary Tree of the BSK Gene Family

In addition to the published literature [13,37,38,39,40], BSK members were retrieved from species such as *Melilotus albus*, *Cicer arietinum*, *Tifgonella pratense*, and *Encalyptus grandisc*, and their evolutionary tree was constructed. The protein sequences of family members were opened with MEGA11 (64-bit) software, and the phylogenetic tree was constructed using the MUSCLE algorithm function with default parameters, using the adjacency method (NJ, Construct/Test Neighbor-Joining Tree). Bootstrap was set to 1000, with other settings being default. Finally, the result was beautified using the Evolview website (http://www.evolgenius.info/evolview, accessed on 6 June 2024).

### 4.4. Chromosome Mapping of the Alfalfa BSK Gene Family

The BSK gene family members, for which position and length information was obtained from the gff3 file, were visualized using TBtools software and renamed based on their distribution on the chromosomes.

### 4.5. Analysis of Collinearity Within and Outside Species

The alfalfa genome files and gff3 files were submitted to the “One Step MCScanX-Super Fast” module, and the generated files were used for mapping, after which the Ka/Ks values were calculated using CDS and protein sequences of the BSK family.

### 4.6. Cis-Acting Elements of Alfalfa BSK Members

The promoter region of BSK genes, spanning 2000 bp upstream, was extracted for analysis on the PlantCARE website (https://bioinformatics.psb.ugent.be/webtools/plantcare/html/, accessed on 15 March 2024), identifying key cis-acting elements. The compiled data were then organized in Excel, and the analytical outcomes were depicted using TBtools for enhanced visualization.

### 4.7. RNA-Seq Analysis of BSK Genes in M. sativa

The downloaded transcriptome data [28] were converted into a SAM file using “Fastq” compression; the data were filtered using “Fastp” and clean reads were kept. The genome of *Medicago sativa‘XinJiangDaYe’* [15] was used as the reference genome, and the average genome comparison rate was 79.25% after indexing with “Hisat2-2.1.0” software. After comparison, the data file format was converted using “Samtools”, the number of genes was calculated with “Stringtie v2.2.1” software to obtain a standardized matrix, and the gene expression level was estimated using the Log_2_ (FPKM + 1) value.

### 4.8. Treatment of M. sativa and qRT-PCR Analysis

The plant material used in this experiment was *M. sativa ‘XinjiangDaYe’*. Seeds were sterilized and planted into seedling pots, using vermiculite as the substrate, and a 1/5 MS nutrient solution was provided via watering every two days; the seedlings were incubated at 22 °C and 75% humidity, with a 16 h light/8 h dark cycle, and treated with salt after 2 weeks. The seedlings were carefully removed from the vermiculite in advance and rinsed with water without damaging the roots. Salt treatments were carried out with three biological replicates of 15 seedlings each at each time point and consisted mainly of immersing the roots of seedlings in a 250 mM NaCl solution for 0, 0.5, 1, 3, 6, and 12 h. The roots were removed, loaded into centrifuge tubes, and quickly frozen using liquid nitrogen to be stored for later use.

Family members with higher expression levels (MsBSK4, MsBSK7, MsBSK9, MsBSK14, MsBSK15, MsBSK16, MsBSK17, MsBSK19, and MsBSK21) were selected for the qRT-PCR assay and used to validate the RNA-seq. Total RNA was extracted using an RNA extraction kit (CWBIO, CW2598S), and RNA concentration was determined using a spectrophotometer. A Golden 1st cDNA Synthesis Kit (HaiGene, D0401) was then utilized for the synthesis of the first strand for a qPCR reaction. The SYBR Green dye method was performed to study the expression patterns of MsBSK genes. The reaction program was conducted as follows: Should be changed to:

95 °C preheating for 5 min, followed by 35 cycles of 95 °C for 20 s, 55 °C for 20 s, 72 °C for 30 s, then 72 °C for 5 min. Three biological replicates were set up for each reaction and their relative expression levels were determined via the 2^−ΔΔCt^ approach. The primer sequences for this study are shown in Appendix A, using the GAPDH gene as the standard control gene.

## 5. Conclusions

In this study, 52 BSK family members were identified in 5 alfalfa subspecies, including 24 MsBSK, 9 MtBSK, 6 MpBSK, 7 MrBSK and 6 MsaBSK genes, and their physicochemical properties were analyzed. Protein sequence lengths ranged from approximately 353 to 526 amino acid residues, and molecular weights ranged from approximately 39,466.97 to 59,700.39 kDa. Alfalfa BSK members were primarily distributed across six branches of the evolutionary tree. Except for the potentially truncated members, MtBSK3-2, MtBSK7-1, and MtBSK7-2, most of the identified BSK members were relatively conserved. BSK genes were unevenly distributed on chromosomes. Intra-species collinearity and Ka/Ks analyses showed that these gene pairs were WGD or segmental, and the genes were subject to purifying selection during evolution. A cis-acting element analysis of alfalfa BSK genes revealed that they contained large numbers of elements related to light response, zein metabolism, endosperm expression, anaerobic induction, drought inducibility, defense and stress, MeJA responsiveness, salicylic acid, and abscisic acid reactions. Transcriptomic profiling under a 250 mM NaCl treatment revealed that the majority of BSK genes reached peak expression at 0.5 h and then declined. Specifically, MsBSK4, MsBSK7, and MsBSK9, which share close relationships with Arabidopsis AtBSK7 and AtBSK8, demonstrated particularly high expression levels at 12 h. Furthermore, these genes displayed significantly higher expression levels in the leaves than in other tissues. The qRT-PCR analysis showed that only the expression trends in the MsBSK4, MsBSK14, MsBSK15, and MsBSK16 genes were consistent with the transcriptome data. Although the expression trends in the MsBSK7 and MsBSK9 genes were generally consistent with the transcriptome data, their expression at 0.5 h–12 h was lower than those of the CK. MsBSK17, MsBSK19, and MsBSK21 registered downregulation at 3 h, which was followed by upregulation. These analyses may provide a foundation for further functional analyses of BSK genes in growth, development, and salt stress tolerance in alfalfa.

## Figures and Tables

**Figure 1 ijms-25-08499-f001:**
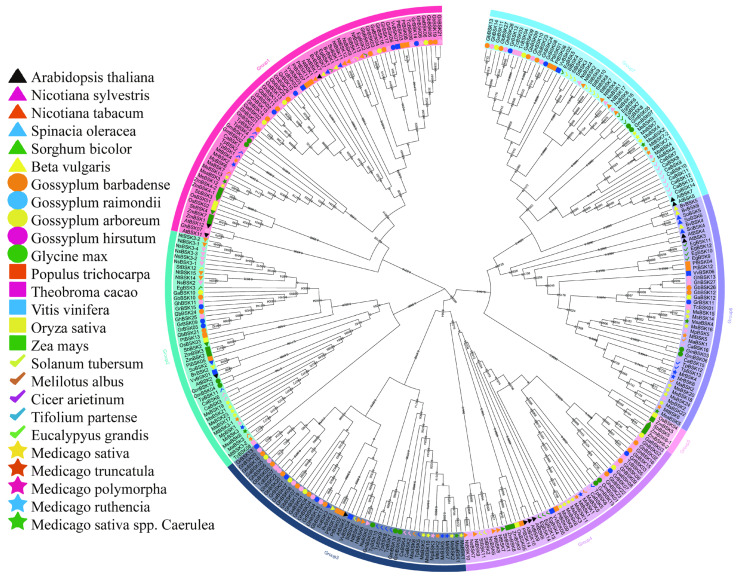
Phylogenetic tree of the BSK gene family. The small number at the fork is the bootstrap score. Phylogenetic trees for multiple species were constructed using MEGA11 with the neighbor-joining (NJ, 1000 bootstraps) method, based on BSK protein sequences. The trees were organized according to evolutionary distance into seven distinct groups, each indicated by a different background color. Additionally, stars of various colors denote the BSK gene families found in the five subspecies of *M. sativa*.

**Figure 2 ijms-25-08499-f002:**
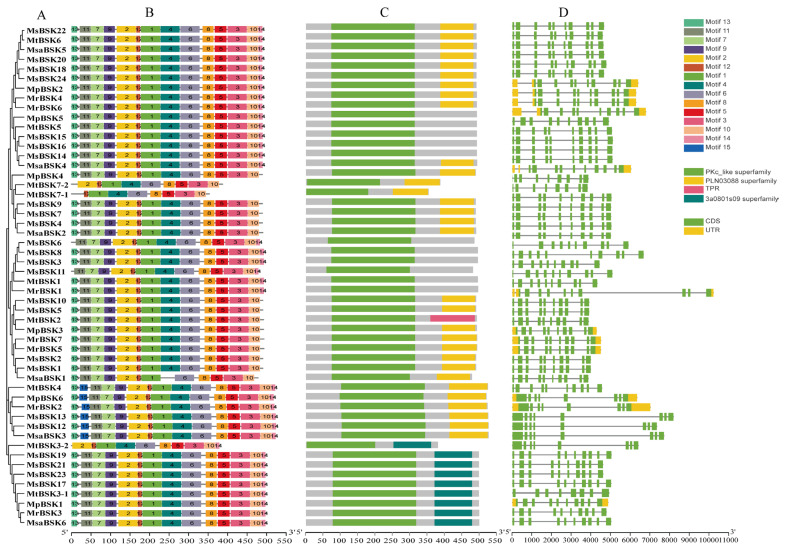
Gene structure, conserved motif, and conserved domain analysis of BSK genes. (**A**) Alfalfa BSK gene family phylogenetic tree. (**B**) BSK gene family conserved motif. (**C**) Domains of the BSK gene family. (**D**) Exon–intron regions of the BSK family.

**Figure 3 ijms-25-08499-f003:**
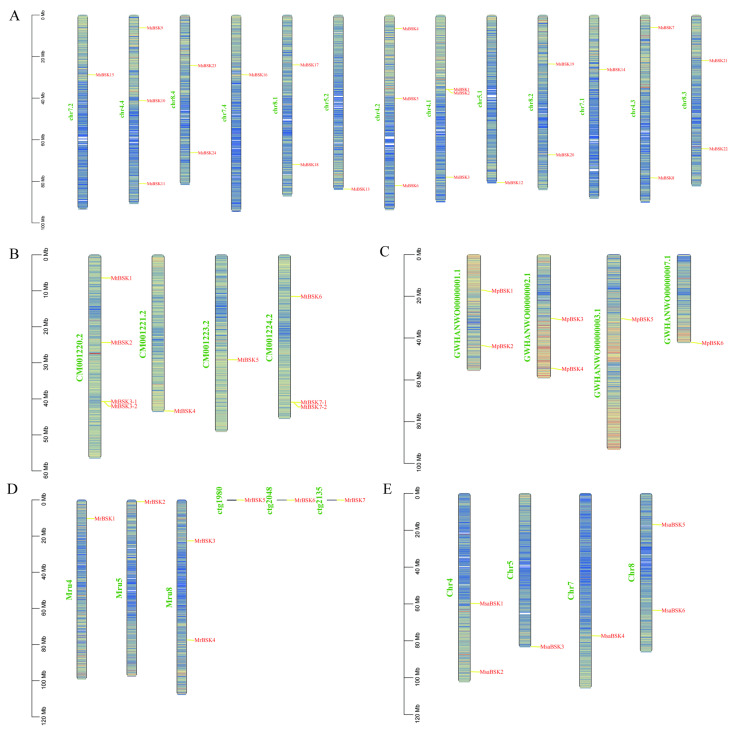
Chromosome localization of five alfalfa BSK families. (**A**–**E**) are the BSK gene mapping of *M. sativa*, *M. truncatula*, *M. polymorpha*, *M. ruthenica,* and *M. sativa* spp. *caerulea*, respectively.

**Figure 4 ijms-25-08499-f004:**
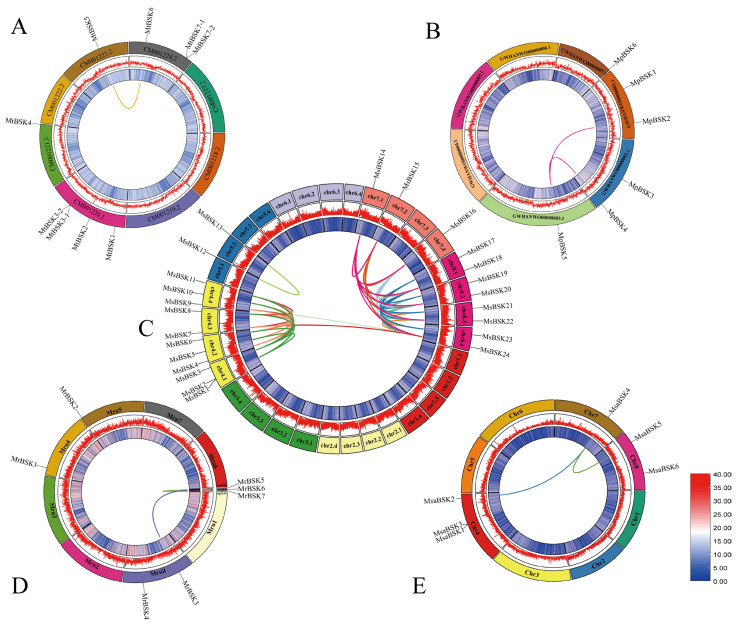
Analysis of collinearity in five species of alfalfa. (**A**) Analysis of intra-collinearity of *M. truncatula*; there is only one syntenic gene pair. (**B**) Collinearity analysis of *M. polymorpha* with two syntenic gene pairs. (**C**) Analysis of intra-collinearity of *M. sativa*; there are forty-two syntenic gene pairs. (**D**) Collinearity analysis of *M. ruthenica* with two syntenic gene pairs. (**E**) Collinearity analysis of *M. sativa* spp. *caerulea* with two syntenic gene pairs. Lower right is gene density.

**Figure 5 ijms-25-08499-f005:**
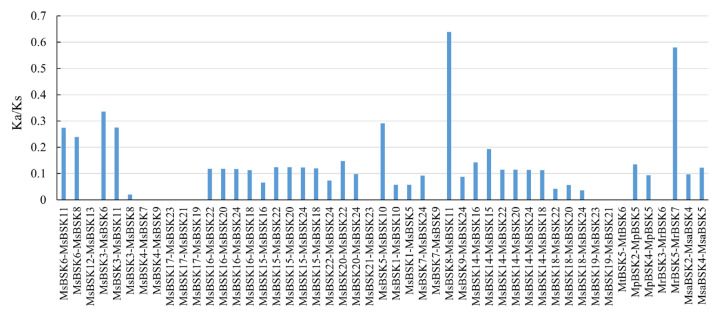
Prediction of Ka/Ks ratios for gene pairs among different species.

**Figure 6 ijms-25-08499-f006:**
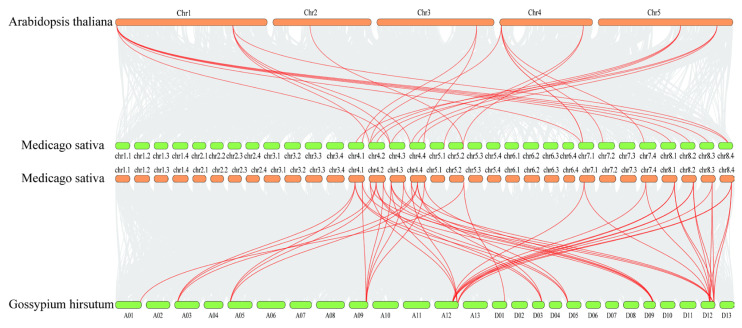
Collinearity analysis of *M. sativa* with *A. thaliana* and *G. hirsutum*. The figure illustrates BSK collinear gene pairs with highlighted red lines and other gene pairs between species indicated by gray areas.

**Figure 7 ijms-25-08499-f007:**
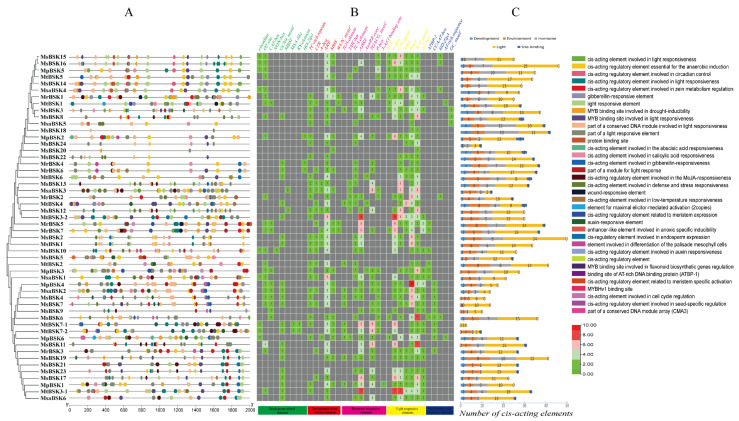
Analysis of BSK gene promoter element of five Alfalfa species. (**A**) Type and number of cis-acting elements in the promoter region. (**B**) Promoter region cis-acting element quantity heat map. (**C**) Number of BSK genes in development, environment, hormone, light, and site binding. The upper-right corner shows the various types of components.

**Figure 8 ijms-25-08499-f008:**
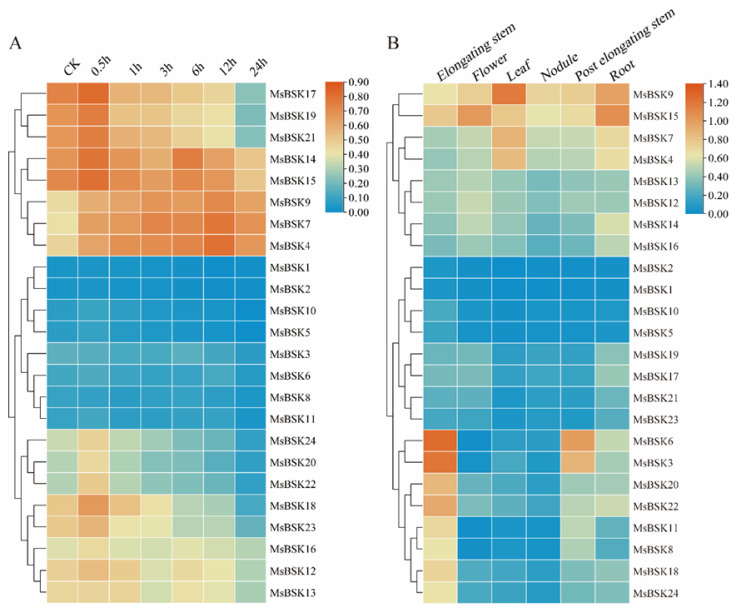
Salt stress and tissue expression heat map of the alfalfa BSK gene family. (**A**) Expression of the MsBSK gene was analyzed after treatment with NaCl for 0, 0.5, 1, 3, 6, 12, and 24 h. (**B**) Expression of the MsBSK gene in six tissues, including elongating stem, flower, leaf, nodule, post-elongating stem, and root.

**Figure 9 ijms-25-08499-f009:**
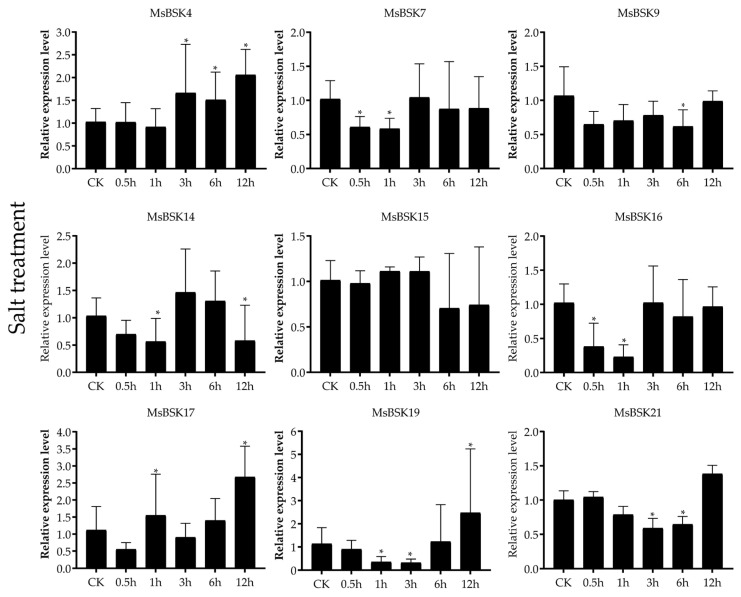
Expression of nine MsBSK genes after 0 h, 0.5 h, 1 h, 3 h, 6 h, and 12 h under salt stress. The mean of three biological replicates was used as the height of the bar graph, and a *p*-value of <0.05 indicated significance, which was marked with *.

**Table 1 ijms-25-08499-t001:** Physicochemical properties of BSK gene family members in five alfalfa species.

Gene ID	Gene Locus	Len.	MW.	pI.	Ins.	AI.	GRAVY	Sub.
MsBSK1	MS.gene055640.t1	490	55,227.43	5.82	38.97	77.08	−0.431	nucl
MsBSK2	MS.gene003974.t1	490	55,227.43	5.82	38.97	77.08	−0.431	nucl
MsBSK3	MS.gene73220.t1	495	55,810.51	6.05	38.51	77.88	−0.388	cyto
MsBSK4	MS.gene72146.t1	489	54,616.92	5.4	42.37	81.23	−0.353	golg
MsBSK5	MS.gene056911.t1	490	55,245.36	5.74	38.91	76.08	−0.453	nucl
MsBSK6	MS.gene92541.t1	485	54,700.26	5.79	38.36	79.9	−0.378	nucl
MsBSK7	MS.gene048802.t1	489	54,616.92	5.4	42.37	81.23	−0.353	golg
MsBSK8	MS.gene037708.t1	495	55,813.51	6.05	40.19	77.88	−0.383	cyto
MsBSK9	MS.gene030398.t1	489	54,616.92	5.4	42.37	81.23	−0.353	golg
MsBSK10	MS.gene30533.t1	490	55,201.31	5.74	38.91	75.9	−0.452	nucl
MsBSK11	MS.gene048125.t1	481	54,367.11	6.69	37.86	79.54	−0.425	nucl
MsBSK12	MS.gene75789.t1	526	59,700.39	6.61	46.51	69.41	−0.654	nucl
MsBSK13	MS.gene65370.t1	525	59,572.26	6.61	46.21	69.54	−0.648	uncle
MsBSK14	MS.gene023870.t1	492	55,145.42	5.3	45.46	81.89	−0.395	cyto
MsBSK15	MS.gene27729.t1	492	55,100.42	5.3	46.15	82.48	−0.381	cyto
MsBSK16	MS.gene52239.t1	492	55,185.53	5.36	47.06	82.48	−0.389	cyto
MsBSK17	MS.gene60383.t1	498	55,888.67	5.91	36.45	81.55	−0.396	cyto
MsBSK18	MS.gene019619.t1	491	55,214.76	5.86	36.37	81.49	−0.396	chlo
MsBSK19	MS.gene011637.t1	498	55,888.67	5.91	36.45	81.55	−0.396	cyto
MsBSK20	MS.gene069204.t1	491	55,244.79	5.86	36.88	81.49	−0.397	chlo
MsBSK21	MS.gene057998.t1	498	55,888.67	5.91	36.45	81.55	−0.396	cyto
MsBSK22	MS.gene071822.t1	491	55,258.81	5.86	36.49	81.69	−0.397	chlo
MsBSK23	MS.gene46356.t1	498	55,888.67	5.91	36.45	81.55	−0.396	cyto
MsBSK24	MS.gene056774.t1	491	55,228.79	5.86	35.92	81.69	−0.396	chlo
MtBSK1	mrna.MTR_4g020070	495	55,812.42	6.01	37.52	77.09	−0.412	nucl
MtBSK2	mrna.MTR_4g065003	492	55,559.82	5.99	36.78	76.95	−0.433	nucl
MtBSK3-1	mrna.MTR_4g098740	498	55,902.7	5.91	36.84	81.55	−0.396	cyto
MtBSK3-2	mrna.MTR_4g098740B	380	42,841.2	5.66	33.17	89.42	−0.232	nucl
MtBSK4	mrna.MTR_5g098970	524	59,447.13	6.55	46.47	70.23	−0.635	nucl
MtBSK5	mrna.MTR_7g077150	492	55,133.36	5.3	46.14	81.28	−0.402	cyto
MtBSK6	mrna.MTR_8g031030	491	55,274.87	5.86	36.53	81.49	−0.392	chlo
MtBSK7-1	mrna.MTR_8g098370C	353	39,466.97	5.59	39.29	84.62	−0.297	golg
MtBSK7-2	mrna.MTR_8g098370D	387	43,262.3	5.41	41.11	85.5	−0.291	golg
MpBSK1	mRNA.Mpo1G14670	498	55,888.67	5.91	36.45	81.55	−0.396	cyto
MpBSK2	mRNA.Mpo1G33860	491	55,214.76	5.86	35.5	81.28	−0.401	cyto
MpBSK3	mRNA.Mpo2G33730	492	55,706.98	5.99	37.06	76.75	−0.439	nucl
MpBSK4	mRNA.Mpo2G12900	489	54,642.9	5.47	40.5	81.02	−0.375	golg
MpBSK5	mRNA.Mpo3G12970	492	55,133.45	5.3	43.83	82.48	−0.384	cyto
MpBSK6	mRNA.Mpo7G0240	520	59,129.78	6.58	46.69	69.46	−0.653	nucl
MrBSK1	evm.model.original_scaffold_314_pilon.25	495	55,872.45	6.1	37.86	76.69	−0.429	nucl
MrBSK2	evm.model.fragScaff_scaffold_221_pilon.305	522	59,458.24	6.56	49.18	70.5	−0.649	nucl
MrBSK3	evm.model.original_scaffold_971_pilon.39	498	55,904.71	5.91	35.6	82.33	−0.384	cyto
MrBSK4	evm.model.fragScaff_scaffold_96_pilon.373	491	55,144.62	5.86	37.58	80.69	−0.407	chlo
MrBSK5	evm.model.original_scaffold_485_pilon.9	492	55,625.79	5.83	39.33	76.95	−0.459	nucl
MrBSK6	evm.model.original_scaffold_708_pilon.13	491	55,144.62	5.86	37.58	80.69	−0.407	chlo
MrBSK7	evm.model.fragScaff_scaffold_48_pilon.47	492	55,587.82	5.91	39.76	75.77	−0.456	nucl
MsaBSK1	MsaT020884.1	477	53,946.9	5.63	37.95	75.89	−0.438	nucl
MsaBSK2	MsaT024189.1	489	54,544.86	5.47	40.65	81.23	−0.346	golg
MsaBSK3	MsaT029913.1	525	59,572.26	6.61	46.21	69.54	−0.648	nucl
MsaBSK4	MsaT038517.1	492	55,207.49	5.3	45.35	81.69	−0.396	cyto
MsaBSK5	MsaT042127.1	491	55,244.79	5.86	36.94	81.49	−0.397	chlo
MsaBSK6	MsaT044678.1	498	55,888.67	5.91	36.45	81.55	−0.396	cyto

Note: Len: numbers of amino acid (aa); MW: molecular weight (kDa); pI: isoelectric point; Ins: instability index; AI: aliphatic index; GRAVY: grand average of hydropathicity; and Sub: subcellular localization.

## Data Availability

The data and materials that support the findings of this study are available from the corresponding authors upon reasonable request.

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
