# Peer review of "Genome-Wide Identification of the Brassinosteroid Signal Kinase Gene Family and Its Profiling under Salinity Stress"

_ijms, 2024, doi:10.3390/ijms25158499_

Round 1

Reviewer 1 Report

Comments and Suggestions for Authors

The authors perform identification  brassinosteroid signal kinase (BSKs) and it expression under salt stress in Medicago family.

The authors do a greta work, perform many detailed analysis etc.

 Hpowever, the text is quaite chaotic, very hard to follow. Moreover, some information are missing.

Namely, I would suggest to „make an order“ in description of different nedicago species and at least in some group information for one Medicag type together.

 Plesae, corrcet all typos and truncated sentences.

Some details:

Line 14: connection require. Please, 1 sentence about BR.

Line 25: “250 mM NaCl stress” -  NaCl treatments or salt stress.

Line 26: „most genes in alfalfa“ ?? Do you mean BSK gene??

Line 35: “Error!”  ¿??

Line 42: “the initial members” ¿?

Line 58: Error..

Line 73- 74: it is nit a good idea to start with supplementary table.

Please, mention figure tuitle only once in legends, do not insert legends in the text.

Line 220: „2.7. Expression Pattern Analysis and qRT-PCR under Tissue and Salt Stress in M. Sativa“ ???? Nothing written about qPCR in the M&M, no details provided.

Line 224 and others: please, do not repeat figure title in tetx. Onc description is enough.

Lines 316 – 317: which alflafa has been used? M.sativa? M.truncatula? Etc?

Line 330-333  truncated sentences.

Lines 357 – 363: where are qPCR description?

Lines 377 – 379: plesae, make uniform.

In many case word are truncated like f and amily on different lines. Please, edit.

Comments on the Quality of English Language

truncated sentences, broken words

Author Response

List of responses:

Major Revision.

Comments: I would suggest to „make an order“ in description of different nedicago species and at least in some group information for one Medicag type together. Plesae, corrcet all typos and truncated sentences.

Response: Thank you for your comments and suggestion. We added species information (xxxxxxx) to the materials and methods, and modified some experimental software and parameters. In addition, whitespace, format, and connectors were also modified.

[The revised manuscript this change can be found –line 370-375.]

Some details:

Comments 1: Line 14: connection require. Please, 1 sentence about BR.

Response 1: Thank you for your comments and suggestion concerning our manuscript. The comments and suggestion are all valuable and very helpful for revising and improving our paper. The revised sentence now reads: " As the sixth class of plant hormones, brassinosteroids (BRs) play indispensable roles in modulating a variety of plant growth, maturation, and environmental adaptation processes, thereby influencing vegetal expansion and development."

[The revised manuscript this change can be found –line 14-16.]

Comments 2: Line 25: “250 mM NaCl stress” - NaCl treatments or salt stress.

Response 2: Thank you for your kind suggestion! We have carefully reviewed the terminology used in the manuscript and have revised the text to replace "250 mM NaCl stress" with " treatment with 250 mM NaCl," to accurately reflect the experimental conditions.

[The revised manuscript this change can be found –line 28.]

Comments 3: Line 26: „most genes in alfalfa“ ?? Do you mean BSK gene??

Response 3: Thank you for your kind suggestion! We have modified the text to ensure that "most genes in alfalfa" unambiguously refers to the alfalfa BSK gene family. The revised sentence now reads: " Expression pattern analysis indicated that the majority of alfalfa genes exhibited downregulation after reaching a peak at 0.5 hours after treatment with 250 mM NaCl, especially for MsBSK14, MsBSK15, MsBSK17, MsBSK19, and MsBSK21; meanwhile, MsBSK4, MsBSK7, and MsBSK9 increased and were highly expressed at 12 hours, demonstrating significantly altered expression patterns under salt stress; furthermore, MsBSK4, MsBSK7, and MsBSK9 exhibited expression specifically in the leaves."

[The revised manuscript this change can be found –line 27-31.]

Comments 4: Line 35: “Error!”  ??

Response 4: Agree. We have identified the error at [1] and[21] have taken immediate action to correct it.

[The revised manuscript this change can be found –line 43,69.]

Comments 5: Line 42: “the initial members”?

Response 5:

Thanks for your suggestion. The use of this term may be inappropriate, so we used " Originally members " in the original text.

Original: The founding members of this family are BSK1 and BSK2, which were identified as BR-responsive proteins by two-dimensional difference gel electrophoresis and mass spectrometry.

Ren H, Willige BC, Jaillais Y, Geng S, Park MY, Gray WM, Chory J. BRASSINOSTEROID-SIGNALING KINASE 3, a plasma membrane-associated scaffold protein involved in early brassinosteroid signaling. PLoS Genet. 2019 Jan 7;15(1):e1007904. doi: 10.1371/journal.pgen.1007904. PMID: 30615605; PMCID: PMC6336344.

[The revised manuscript this change can be found –line 50.]

Comments 6: Line 58: Error

Response 6: Thank you for your kind suggestion! Agree. We have identified the error at [1] and[21] have taken immediate action to correct it.

[The revised manuscript this change can be found –line 69.]

Comments 7: Line 73- 74: it is nit a good idea to start with supplementary table.Please, mention figure tuitle only once in legends, do not insert legends in the text.

Response 7: Thank you for your kind suggestion! We fixed this issue and changed "Table S1" to "Table 1", and cancelled the insertion.

[The revised manuscript this change can be found –line 83,100.]

Comments 8: Line 220: „2.7. Expression Pattern Analysis and qRT-PCR under Tissue and Salt Stress in M. Sativa“ ???? Nothing written about qPCR in the M&M, no details provided.

Response 8: Thank you for your kind suggestion! We have added "2.8.qRT-PCR under Salt Stress in M. sativa" to the latest version of the modification and have described this part.

[The revised manuscript this change can be found –line 269-284.]

Comments 9: Line 224 and others: please, do not repeat figure title in tetx. Onc description is enough.

Response 9: Thank you for your kind suggestion! We fixed this issue by removing the insertion of "Figure 8-A" and "Figure 8-B" and entering the text format instead.

[The revised manuscript this change can be found –line 244,254.]
Comments 10: Lines 316 – 317: which alflafa has been used? M.sativa? M.truncatula? Etc?

Response 10: Thank you for your kind suggestion! Five types of alfalfa are used here, namely "M. sativa, M. truncatula, M. polymorpha, M. ruthenica, M. sativa spp. caerulea". And added” (M.sativa, ‘XinJiangDaYe’, 2n = 4× = 32; M.truncatula, MedtrA17_4.0, 2n = 2x = 16; M.polymorpha, GWHANWO00000000.1, 2n = 2x = 14; M.ruthenica, ASM1820801v1, 2n = 2x = 16; M. sativa spp. caerulea, PI464715; 2n = 2x = 16)”

[The revised manuscript this change can be found –line 371-373.]

Comments 11: Line 330-333  truncated sentences.

Response 11: Thank you for your kind suggestion! To solve this problem, we selected the "Unhyphenate" option.

[The revised manuscript this change can be found –line 383-387.]

Comments 12: Lines 357 – 363: where are qPCR description?

Response 12: Agree. We added: "4.8. Treatment of M. sativa and qRT-PCR analysis" in the method section and described the experimental content in detail.

[The revised manuscript this change can be found –line419-442.]

Comments 13: Lines 377 – 379: plesae, make uniform.

Response 13: Thank you for your kind suggestion! We have corrected this part of the error.

[The revised manuscript this change can be found –line454-457.]

Comments 14: In many case word are truncated like f and amily on different lines. Please, edit.

Response 14: Thank you for your kind suggestion! This was previously set up to adjust for word spacing issues, which we have fixed.

Reviewer 2 Report

Comments and Suggestions for Authors

This MS has solid and comprehensive scientific content, and its results can contribute to the current knowledge in genetics and salt stress resistance in alfalfa. However, it needs significant modifications in various parts. After these corrections are made, the article can be accepted for publication.

-         The abstract of the article covers general information and main results well. However, it is suggested that more details of the key results, such as the number of identified genes and changes in gene expression due to salt stress, be provided.

-         The introduction is generally well-structured, providing a comprehensive overview of BRs, their roles, and the significance of the BSK gene family. However, the paragraph could benefit from a more precise separation between general information about BRs and specific details about the BSK gene family.

-         Some sentences are pretty complex and could be simplified for better readability. For example, breaking down the sentence on the roles and mechanisms of BSK family proteins in alfalfa into smaller parts would enhance clarity.

-         In M&M, it would be better to provide more details about the exact parameters of the experiments and the software used.

-         results are generally comprehensive and complete, but some can be presented briefly to avoid repetition.

-         discussion has been well interpreted and compared the results with previous research. Important points about the effect of BSK genes in response to salt stress have been stated. However, some arguments require stronger arguments.

-         For more comment, plz check uploaded file

-         The entire text needs a general revision to fix grammatical and writing errors.

Comments on the Quality of English Language

Extensive editing of English language required

Author Response

 Comments 1: The abstract of the article covers general information and main results well. However, it is suggested that more details of the key results, such as the number of identified genes and changes in gene expression due to salt stress, be provided.

Response 1: Thank you for pointing this out. The number of genes identified in the question was including 24 MsBSK, 9 MtBSK, 6 MpBSK, 7 MrBSK, and 6 MsaBSK genes, a total of 52 genes were identified in 5 species of alfalfa. In addition, changes in gene expression were seen in transcriptome data and qRT-PCR.

[Location of this change in the revised draft – lines 82,253-262.]

Comments 2: The introduction is generally well-structured, providing a comprehensive overview of BRs, their roles, and the significance of the BSK gene family. However, the paragraph could benefit from a more precise separation between general information about BRs and specific details about the BSK gene family.

Response 2: Thank you for pointing this out. We've noticed the problem and have made the change you suggested to include the sentence in that location: Brassinosteroids (BRs), recognized as the sixth major class of plant hormones, play crucial roles in various aspects of plant growth and development, including seed germination, cell elongation, photomorphogenesis, xylem differentiation, and reproduction [1]. Not only do they contribute to normal plant development, but they also assist in resisting high-salinity and alkaline conditions. Exogenous application of BRs has been shown to bolster seed resistance to salinity stress [2,3,4,5,6], while also alleviating the harmful effects of saline conditions and alkali on plant health, according to studies [7,8]. BR signal kinase (BSK) is an important cytoplasmic receptor kinase downstream of the BR signal transduction pathway that serves as a substrate of brassinosteroid receptor BRI [9] and plays a crucial role in regulating plant growth, development, and abiotic stress responses.

[Location of this change in the revised draft – lines 40-49.]

Comments 3: Some sentences are pretty complex and could be simplified for better readability. For example, breaking down the sentence on the roles and mechanisms of BSK family proteins in alfalfa into smaller parts would enhance clarity.

Response 3: Thank you for pointing this out. We rewrote the sentence: BR signal kinase (BSK) is an important cytoplasmic receptor kinase downstream of the BR signal transduction pathway that serves as a substrate of brassinosteroid receptor BRI [9] and plays a crucial role in regulating plant growth, development, and abiotic stress responses.

[Location of this change in the revised draft – lines 47-49.]

Comments 4: In M&M, it would be better to provide more details about the exact parameters of the experiments and the software used.

Response 4: Thank you for pointing this out. The software and parameters have been described in the original text, and we have also added species information: (M.sativa, ‘XinJiangDaYe’, 2n = 4× = 32; M.truncatula, MedtrA17_4.0, 2n = 2x = 16; M.polymorpha, GWHANWO00000000.1, 2n = 2x = 14; M.ruthenica, ASM1820801v1, 2n = 2x = 16; M. sativa spp. caerulea, PI464715; 2n = 2x = 16).

[Location of this change in the revised draft – lines 371-373.]

Comments 5: results are generally comprehensive and complete, but some can be presented briefly to avoid repetition.

Response 5: Thank you for pointing this out. We have realigned part of the description of the results and modified part of the content.

[Location of this change in the revised draft – lines 444-470.]

Comments 6: discussion has been well interpreted and compared the results with previous research. Important points about the effect of BSK genes in response to salt stress have been stated. However, some arguments require stronger arguments.

Response 6: Thank you for pointing this out. In the current relevant literature, only the genes AtBSK3, AtBSK5, and ZmBSK7 have been investigated to be associated with salt stress, which is also described in the original text.

[Location of this change in the revised draft – lines 299-309.]

In Article

Comments 1: Line 13:Alfalfa(Medicago L.)

Response 1: Thank you for pointing this out. We adjusted the spacing and made changes in the article.

[The revised manuscript this change can be found –line 12.]

Comments 2: Line 17: in 5 alfalfa species.

Response 2: Thank you for pointing this out. We changed "5" to "five"

[The revised manuscript this change can be found –line 19.]

Comments 3: Line 17: have been rarely.

Response 3: Thank you for pointing this out. We changed " have been rarely " to " have rarely been "

[The revised manuscript this change can be found –line 19.]

Comments 4: Line 18: from.

Response 4: Thank you for pointing this out. We changed " from the genomes in " to " in the genomes in "

[The revised manuscript this change can be found –line 20.]

Comments 5: Line 18: 5 species

Response 5: Thank you for pointing this out. We changed " from the genomes in " to " in the genomes in "

[The revised manuscript this change can be found –line 20.]

Comments 6: Line 22: BSK genes were influence by Purifying Selection.

Response 6: Thank you for pointing this out. We changed " BSK genes were influence by Purifying Selection " to " Purifying Selection influenced BSK genes "

[The revised manuscript this change can be found –line 25.]

Comments 7: Line 28: a

Response 7: Thank you for pointing this out. We have joined: a.

[The revised manuscript this change can be found –line 35.]

Comments 8: Line 29: of

Response 8: Thank you for pointing this out. We changed " of " to " in "

[The revised manuscript this change can be found –line 36.]

Comments 9: Line 41: the regulation of

Response 9: Thank you for pointing this out. We changed " the regulation of " to "regulating"

[The revised manuscript this change can be found –line 49.]

Comments 10: Line 52 : very important

Response 10: Thank you for pointing this out. We changed " very important " to " great significance  "

[The revised manuscript this change can be found –line 59.]

Comments 11: Line 55: biomass of livestock

Response 11: Thank you for pointing this out. We changed "biomass of livestock " to " livestock biomass "

[The revised manuscript this change can be found –line 63.]

Comments 12: Line 60: and

Response 12: Thank you for pointing this out. We added "and"

[The revised manuscript this change can be found –line 71.]

Comments 13: Line 67-68:

Response 13: Thank you for pointing this out. We have revised this sentence: Building on previous findings, the post-treatment expression and qRT-PCR data of alfalfa were further explored, providing theoretical insights into the function of BSK genes in response to salinity stress.

[The revised manuscript this change can be found –line 63.]

Reviewer 3 Report

Comments and Suggestions for Authors

In the article “Identification of BSK Gene Family in Alfalfa and Its Response to Salt Stress”, the authors analyzed BSK gene family that is related to Brassinosteroid (BR) signaling, as BR played important roles in abiotic stress tolerance authors also analyzed the gene expression profiles under salinity stress. However, I order to publish the paper in IJMS, it needs to be improved in terms of many aspects. Here I have provided my detailed comments, and suggests major revision, which can be addressed before publication.

Major Problems

1.      There are many typing mistakes in the manuscript,  and it seems that authors didn’t proofread the article before submission e.g. “[Error! Reference source not found.]” in line 35 and 58, moreover at many places the authors didn’t provide appropriate space between words like in line 13, line 79, line 80 and at many other places.

2.      The English quality is not up to the mark, at many places the sentence structure is not appropriate, so I would suggest English editing from a native English speaker.

3.      Another problem is the novelty and content of the paper, as the paper only identifies the gene family, so there must be more data, like I would suggest to add gene expression profiling under various other stresses as well, like drought, heavy metal, heat stress etc. I am quite sure, there must be some RNA seq data available from where the authors can get the gene expression data.

4.      Another major problem is that the legends  of the figures are very brief, so provide detailed legends for each figures, which explains that what the figures is going to tell the readers.

5.      Also change the word salt-alkali stress to salinity stress throughout the manuscript.

6.      The plagiarism must be below 20 percent, as the plagiarism is 21% in current version of manuscript.

Other Minor Issues

Title

Comment 0: The title can be changed to “Genome wide identification of BSK gene family and its profiling under salinity stress

Abstract:

Comment 1: Line 13-14:  Rewrite the lines as it is not understandable “has the advantage of ………………soil improvement” if you are focusing only salt stress, then there is no need to write for soil improvement as well.

Comment 2: Lines 23-25: Rewrite the following lines because they are very poorly written and not understandable, so write them in an logical manner “Most of the cis-acting………….abscisic acid”.

Comment 3: Line 25: Change “Analysis of expression to “expression analysis”.

Comment 4: Line 26: Change “a downward trend” to “downregulation”

Comment 5: Line 13: Provide space between Alfalfa and (Medicago L.).

Introduction

Comment 6: Line 33: Change the line as “Brassinosteroid (BR) as a sixth major plant hormone, plays an important role in………”

Comment 7: Line 35: Please correct the citation in your reference manager “[Error! Reference source not found.]”

Comment 8: The introduction lacks information related to salinity stress and its impact on Alfalfa crop, so provide at least one paragraph of 3-4 lines which explains that how salinity impacts Alfalfa crop and why it is very important stress for this crop.

Results

Comment 9: Verify if the AtBSK needs to be italic or not.

Comment 10: Figure 1: The phylogenetic tree is very complex in its current form, and very difficult to read as the authors added a lot crops it. I would suggest to add only Arabidopsis, Nicotiana tabacum, oryza sativa, all Medicago species. So, I would be more beautiful and will be easy for the readers to understand.

Comment 11: Figure 3: The figure is in very bad quality and not readable, please change or modify it, so the readers can easily read.

Comment 12: Figure 4: Same is the case with collinearity analysis, first of all provide a, b, c marking over the figures, secondly make it more visible and clear.

Comment 13: Figure 6: Explain in figure legend and in results as well, that which Medicago specie was used for collinearity analysis in this figure, and why other were not used.

Comment 14: Figure 7: Once again very complex figure, make the figure beautiful and make it logical for the understanding of readers. In it current form, its very difficult to read on 100% zoom.

Comment 15: Figure 8: on which basis the genes for salt stress expression were selected? Moreover I suggest to add the expression under other abiotic stresses as well.

Comments on the Quality of English Language

Author Response

Major Problems

Comments 1: There are many typing mistakes in the manuscript, and it seems that authors didn’t proofread the article before submission e.g. “[Error! Reference source not found.]” in line 35 and 58, moreover at many places the authors didn’t provide appropriate space 6between words like in line 13, line 79, line 80 and at many other places.

Response 1: Thank you for pointing this out. We've reinserted the document source.

[The revised manuscript this change can be found –line 43,69.]

Comments 2: The English quality is not up to the mark, at many places the sentence structure is not appropriate, so I would suggest English editing from a native English speaker.

Response 2: Thank you for pointing this out. We take your suggestions and look for journals to recommend English touch-up sites for revisions.

Comments 3: Another problem is the novelty and content of the paper, as the paper only identifies the gene family, so there must be more data, like I would suggest to add gene expression profiling under various other stresses as well, like drought, heavy metal, heat stress etc. I am quite sure, there must be some RNA seq data available from where the authors can get the gene expression data.

Response 3: Thank you for pointing this out. In the whole article, because we only studied the salt stress of this gene family, it is not suitable to add RNA-seq data for other abiotic stresses. Therefore, only the registration numbers of: SRR7160314-SRR7160315, SRR7160322-SRR7160331, SRR7160339-SRR7160341, SRR7160351-SRR7160352 and SRR7160354-SRR7160357; SRR1823816-SRR1823833 were downloaded from NCBI. These data were aligned with ‘XinJiangDaYe’ as the reference genome. The final PFKM values were calculated to plot a heat map for the display of gene expression.

[The revised manuscript this change can be found –line 411-418.]

Comments 4: Another major problem is that the legends of the figures are very brief, so provide detailed legends for each figures, which explains that what the figures is going to tell the readers.

Response 4: Thank you for your suggestion. We have included details in each legend to make it easier for the reader to better understand. For the image sharpness issue, we redrew the image and adjusted the dpi value to 600.

[The revised manuscript this change can be found –line 122,147,172,185,201,213,235,265 and 282.]

Comments 5: Also change the word salt-alkali stress to salinity stress throughout the manuscript.

Response 5: Thank you for pointing this out. We have amended this in the appropriate places.

[The revised manuscript this change can be found –line 13.]

Comments 6: The plagiarism must be below 20 percent, as the plagiarism is 21% in current version of manuscript. After ‘iThenticate’ check, the check rate was 19%.

Response 6: Thank you for pointing this out. We re-checked the article and revised it according to the marks in the check report to ensure that the check rate compounded the requirements.

Other Minor Issues

Title

Comments 0: The title can be changed to “Genome wide identification of BSK gene family and its profiling under salinity stress

Response 0: Thank you for pointing this out. I agree with this comment. Therefore, I accept your suggestion and modify it accordingly.

[The revised manuscript this change can be found –line 2.]

Abstract:

Comments 1: Line 13-14: Rewrite the lines as it is not understandable “has the advantage of ………………soil improvement” if you are focusing only salt stress, then there is no need to write for soil improvement as well.

Response 1: Thank you for pointing this out. Here is just a brief overview of some of the benefits of alfalfa. We reworked the sentence as: Alfalfa (Medicago L.), is a high-quality perennial leguminous forage with the advantages of salt tolerance; its mowing tolerance and high protein content render it the most economically valuable forage.

[The revised manuscript this change can be found –line 12-13.]

Comments 2: Lines 23-25: Rewrite the following lines because they are very poorly written and not understandable, so write them in an logical manner “Most of the cis-acting………….abscisic acid”.

Response 2: Thank you for pointing this out. We reworded the sentence to read: most of the cis-acting elements in the promoter region were associated with responses such as light, defense and stress, anaerobic induction, MeJA, and abscisic acid.

[The revised manuscript this change can be found –line 26-27.]

Comments 3: Line 25: Change “Analysis of expression to “expression analysis”.

Response 3: Thank you for pointing this out. We reworded the sentence to read: Expression pattern analysis

[The revised manuscript this change can be found –line 27.]

Comments 4: Line 26: Change “a downward trend” to “downregulation”

Response 4: Thank you for pointing this out. We have made modifications based on your suggestions.

[The revised manuscript this change can be found –line 28.]

Comments 5: Line 13: Provide space between Alfalfa and (Medicago L.).

Response 5: Thank you for pointing this out. We adjusted the spacing and made changes in the article.

[The revised manuscript this change can be found –line 12.]

Introduction

Comments 6: Line 33: Change the line as “Brassinosteroid (BR) as a sixth major plant hormone, plays an important role in………”

Response 6: Thank you for pointing this out. We modify the sentence to read: Brassinosteroids (BRs), recognized as the sixth major class of plant hormones, play crucial roles in various aspects of plant growth and development, including seed germination, cell elongation, photomorphogenesis, xylem differentiation, and reproduction [1].

[The revised manuscript this change can be found –line 40-43.]

Comments 7: Line 35: Please correct the citation in your reference manager “[Error! Reference source not found.]”

Response 7: Thank you for pointing this out. We have re-cited the literature in this position. [The revised manuscript this change can be found –line 43,69.]

Comments 8: The introduction lacks information related to salinity stress and its impact on Alfalfa crop, so provide at least one paragraph of 3-4 lines which explains that how salinity impacts Alfalfa crop and why it is very important stress for this crop.

Response 8: Thank you for your suggestion! The newly added sentence is: High-salt environments induce osmotic stress and ion toxicity in alfalfa, leading to stunted growth and reduced biomass. Consequently, understanding the mechanisms of salt tolerance in alfalfa and developing effective salt tolerance strategies are essential to mitigatinge the adverse effects of salinity stress on this crop.

[The revised manuscript this change can be found –line 64-67.]

Results

Comments 9: Verify if the AtBSK needs to be italic or not

Response 9: Thank you for your suggestion! The convention is to italicize the abbreviations of gene names when they are used to refer to the gene itself. Therefore, "AtBSK" should be italicized when it is first mentioned to indicate that it is a gene name. However, the use of "AtBSK" in the following text is mainly used to refer to protein sequences, so italics are no longer used.

[The revised manuscript this change can be found –line 50.]

Comments 10: Figure 1: The phylogenetic tree is very complex in its current form, and very difficult to read as the authors added a lot crops it. I would suggest to add only Arabidopsis, Nicotiana tabacum, oryza sativa, all Medicago species. So, I would be more beautiful and will be easy for the readers to understand.

Response 10: Thank you for your suggestion! We have noted the complexity of the chart and have detailed in the Figure 1 section how to interpret the evolutionary tree and how to identify key evolutionary branches and nodes. Using the BSK gene family of multiple species allows us to show the distribution and diversity of the gene family in different species, which helps to more accurately infer the evolutionary relationships between gene family members and to understand the conservation and specificity of these genes during evolution.

[The revised manuscript this change can be found –line 122-127.]

Comments 11: Figure 3: The figure is in very bad quality and not readable, please change or

modify it, so the readers can easily read.

Response 11: Thank you for your suggestion! In response to your comment, we have taken the following steps to improve the readability of Figure 3: We redrew the graph and increased the image resolution to 300dpi to ensure that all elements are legible.

[The revised manuscript this change can be found –line 171-173.]

Comments 12: Figure 4: Same is the case with collinearity analysis, first of all provide a, b, c marking over the figures, secondly make it more visible and clear.

Response 12: Thanks for your suggestion! We reworked the figure, adjusted the dpi of the image, and added an explanation to the legend.

[The revised manuscript this change can be found –line 185-189.]

Comments 13: Figure 6: Explain in figure legend and in results as well, that which Medicago specie was used for collinearity analysis in this figure, and why other were not used.

Response 13: Thanks for your suggestion! We have included in this paragraph the reasons for the selection of A. thaliana and G. hirsutum, with information on the selected species in the previous legend.

[The revised manuscript this change can be found –line 202-205.]

Comments 14: Figure 7: Once again very complex figure, make the figure beautiful and make it logical for the understanding of readers. In it current form, its very difficult to read on 100% zoom.

Response 14: Thanks for your suggestion! The graph is mainly divided into three parts, namely the cis-acting element diagram, the number of cis-acting elements heat map, and the number of BSK genes in development, environment, hormones, light and check point binding. We have added an explanation below the picture for the reader to better understand.

[The revised manuscript this change can be found –line 235-239.]

Comments 15: Figure 8: on which basis the genes for salt stress expression were selected? Moreover I suggest to add the expression under other abiotic stresses as well.

Response 15: Thanks for your suggestion. The genes shown in Image 8 are the expression of the MsBSK gene family obtained by comparing the transcriptome data with the reference genome of M.sativa "XinJiangDaYe". On this basis, we selected the genes with higher expression for qRT-PCR, which will be shown in the next paragraph. In addition, the whole article is mainly about salt stress, so it is not appropriate to include other abiotic stress RNA-seq data. Some notes have also been added to the figure.

[The revised manuscript this change can be found –line 265-268.]

Round 2

Reviewer 1 Report

Comments and Suggestions for Authors

Thank you for the great work! Now the text is almost ready. Small comments:

Line 101: numbers of aminoacid.

In addition, extensive  proof reading are required.

Next time, please, write you respons ein Engish, not in China.

Comments on the Quality of English Language

proof reading

Author Response

Comments 1: Thank you for the great work! Now the text is almost ready. Small comments: Line 101: numbers of aminoacid. In addition, extensive  proof reading are required. Next time, please, write you respons ein Engish, not in China.

Response 1: Thank you for the good comment. We changed "length" to "numbers" as required.

[The revised manuscript this change can be found –line 101.]

Comments 2: Line 97,.

Response 2: Agree. We provide spaces at "," and "Subcellular"

[The revised manuscript this change can be found –line 97.]

Comments 3:Line 133.

Response 3: Thank you for the good comment. We replaced "feom" with "from".

[The revised manuscript this change can be found –line 133.]

Comments 4:Line 182.

Response 4: Thank you for the good comment. We replaced "genome-wide or segmental duplicates" with "WGD or Segmental”.

[The revised manuscript this change can be found –line 182.]

Comments 5:Line 203.

Response 5: Thank you for the good comment. We'll make it italic: A. thaliana.

[The revised manuscript this change can be found –line 203.]

Comments 6:Line 249.

Response 6: Thank you for the good comment. We changed the sequence of gene names.

[The revised manuscript this change can be found –line 249.]

Comments 7:Line 293.

Response 7: Thank you for the good comment. We'll make it italic: A. thaliana.

[The revised manuscript this change can be found –line 293.]

Comments 8:Line 301.

Response 8: Thank you for the good comment. We'll make it italic: Pseudomonas syringae.

[The revised manuscript this change can be found –line 301.]

Comments 9:Line 302.

Response 9: Thank you for the good comment. We'll make it italic: Botrytis cinerea.

[The revised manuscript this change can be found –line 302.]

Comments 10:In line 316,327,329.

Response 10: Thank you for the good comment. We added a comma.

[The revised manuscript this change can be found –line 316,327,329.]

Comments 11:Line 419.

Response 11: Thank you for the good comment. We'll make it italic: M. sativa ‘XinjiangDaYe.

[The revised manuscript this change can be found –line 419.]

Comments 12:Line 452.

Response 12: Thank you for the good comment. We replaced "genome-wide or segmental duplicates" with "WGD or Segmental”.

[The revised manuscript this change can be found –line 452.]

Reviewer 2 Report

Comments and Suggestions for Authors

With the changes made, the article can be accepted in its current form.

Author Response

Dear Reviewer 2,  

Thank you so much for your diligent review and constructive feedback on our manuscript titled " Identification of BSK Gene Family in Alfalfa and Its Response to Salt Stress " with the submission ID: ijms-3107603. We greatly appreciate the time and effort you have dedicated to evaluating our work.  

Best regards,

Biao Shi

Reviewer 3 Report

Comments and Suggestions for Authors

The paper could be accepted, as all the raised concerned are removed by author.

Author Response

(The authors gave the same response as above.)
